# Precocious Downregulation of Krüppel-Homolog 1 in the Migratory Locust, *Locusta migratoria*, Gives Rise to An Adultoid Phenotype with Accelerated Ovarian Development but Disturbed Mating and Oviposition

**DOI:** 10.3390/ijms21176058

**Published:** 2020-08-22

**Authors:** Marijke Gijbels, Sam Schellens, Tine Schellekens, Evert Bruyninckx, Elisabeth Marchal, Jozef Vanden Broeck

**Affiliations:** 1Research Group of Molecular Developmental Physiology and Signal Transduction, KU Leuven, Zoological Institute, Naamsestraat 59 box 2465, 3000 Leuven, Belgium; Marijke.Gijbels@kuleuven.be (M.G.); Sam.Schellens@kuleuven.be (S.S.); Tine_schellekens@hotmail.com (T.S.); Evert.Bruyninckx@kuleuven.be (E.B.); 2Life Science Technologies, Imec, Kapeldreef 75, B- 3001 Leuven, Belgium

**Keywords:** ecdysteroid, Hemimetabola, insect, juvenile hormone, metamorphosis, reproduction

## Abstract

Krüppel-homolog 1 (Kr-h1) is a zinc finger transcription factor maintaining the status quo in immature insect stages and promoting reproduction in adult insects through the transduction of the Juvenile Hormone (JH) signal. Knockdown studies have shown that precocious silencing of *Kr-h1* in the immature stages results in the premature development of adult features. However, the molecular characteristics and reproductive potential of these premature adult insect stages are still poorly understood. Here we report on an adult-like or ‘adultoid’ phenotype of the migratory locust, *Locusta migratoria*, obtained after a premature metamorphosis induced by the silencing of *LmKr-h1* in the penultimate instar. The freshly molted adultoid shows precocious development of adult features, corresponding with increased transcript levels of the adult specifier gene *LmE93*. Furthermore, accelerated ovarian maturation and vitellogenesis were observed in female adultoids, coinciding with elevated expression of *LmCYP15A1* in *corpora allata* (CA) and *LmKr-h1* and *vitellogenin* genes (*LmVg*) in fat body, whereas *LmE93* and *Methoprene-tolerant* (*LmMet*) transcript levels decreased in fat body. In adultoid ovaries, expression of the *Halloween* genes, *Spook* (*LmSpo*) and *Phantom* (*LmPhm*), was elevated as well. In addition, the processes of mating and oviposition were severely disturbed in these females. *L. migratoria* is a well-known, swarm-forming pest insect that can destroy crops and harvests in some of the world’s poorest countries. As such, a better understanding of factors that are capable of significantly reducing the reproductive potential of this pest may be of crucial importance for the development of novel locust control strategies.

## 1. Introduction

The transition from immature insect stages to fully reproductive adults is dependent on two hormones, i.e., Juvenile Hormone (JH) and ecdysteroids. Whereas periodic pulses of ecdysteroids trigger the molting process in juvenile insects, the absence or presence of JH will determine the nature of the molt [1]. JH is a sesquiterpenoid hormone synthesized in the *corpora allata* (CA), which are part of the retrocerebral complex of insects. In addition to their role in immature insect stages, both ecdysteroids and JH are endocrine regulators of reproduction in adult insects. Ecdysteroids induce their effects by binding to a heterodimeric complex of two nuclear receptors, the ecdysone receptor (EcR) and the retinoid-X-receptor/ultraspiracle (RXR/USP), as reviewed by Hill et al. (2013) [2]. The JH signaling pathway was only recently described with the *in vitro* and *in vivo* characterization of a genuine JH receptor, Methoprene-tolerant (Met). Met is a transcription factor belonging to the basic helix–loop–helix (bHLH)/Per-Arnt-Sim (PAS) family [3,4]. The binding of JH stimulates Met to form a complex with other bHLH-PAS proteins such as Taiman (Tai), a steroid receptor co-activator. The JH-Met-Tai complex then induces the expression of JH response genes such as *Krüppel-homolog 1* (*Kr-h1*), the main effector of the anti-metamorphic action of JH [5,6]. In a variety of insect species, RNA interference (RNAi) studies have demonstrated the importance of Met and its downstream transcription factor Kr-h1 in the anti-metamorphic effects of JH. Knockdown of either *Met* or *Kr-h1* in larvae of the red flour beetle, *Tribolium castaneum*, induced a precocious metamorphosis to pupae [7,8,9]. Reduced *Met* or *Kr-h1* transcript levels in nymphs of the German cockroach, *Blattella germanica*, the kissing-bug, *Rhodnius prolixus*, the cricket, *Gryllus bimaculatus*, the brown planthopper, *Nilaparvata lugens*, the common bed bug, *Cimex lectularius*, and the fire bug, *Pyrrhocoris apterus*, resulted in precocious development of adult features [10,11,12,13,14,15,16,17]. Although the occurrence of a precocious metamorphosis resulting in adultoid insects is well known, its impact on reproductive physiology is only poorly understood. Nowadays, only the morphology of reproductive organs has been described in adultoids of a few species. In adultoids of the bed bug, *C. lectularius*, induced by RNAi-mediated knockdown of *Kr-h1,* precocious development of ovaries and oviducts was observed [17]. In addition, external genitals were found in *dsKr-h1*-injected *N. lugens* and *P. apterus* [10,16,18], while an aberrant ovipositor morphology was described in adultoid crickets of the species *G. bimaculatus* [15].

While Kr-h1 mediates the status quo action of JH, the ecdysteroid early response gene, E93, is a key determinant promoting adult morphogenesis in both holo- and hemimetabolan insects. In nymphs of *B. germanica*, *G. bimaculatus*, *C. lectularius* and *N. lugens*, E93 depletion prevented the nymphal-to-adult transition, giving rise to supernumerary nymphal instars [15,16,17,19]. In the holometabolan insect *T. castaneum*, E93 depletion in larvae prevented the larval-to-pupal or pupal-to-adult transition, resulting in the formation of a supernumerary second pupa or inducing reiteration of larval development depending on the time of double-stranded (ds)RNA injection [9,19]. These studies have established the inhibitory effect of Kr-h1 on E93 expression and *vice versa* [9,19,20]. This ancestral regulatory mechanism by which JH inhibits metamorphosis is now defined as the MEKRE93 axis. During nymphal-to-nymphal transitions, high JH titers induce *Kr-h1* expression (via Met), which in turn represses the expression of *E93*. In the final juvenile stage, when JH titers drop, *Kr-h1* expression is interrupted, resulting in a strong induction of E93 through ecdysone signaling, which triggers adult development (as reviewed by Belles, 2019) [21].

Moreover, in adult insects, JH transduces its signal via Met and Kr-h1. Multiple studies have confirmed the involvement of JH in vitellogenin synthesis in the fat body, lipid accumulation in the primary oocyte, regulation of mating and sex pheromone production in several insects [22,23,24,25,26,27,28,29,30,31,32,33,34,35,36]. Moreover, an effect on the number of eggs deposited could be attributed to JH as well, since a reduced number of eggs laid was observed upon silencing *Met* or *Kr-h1* in the brown planthopper, *N. lugens* [22]. Furthermore, in locusts, such as *L. migratoria*, RNAi-studies have shown the importance of *Met* and *Kr-h1* in female reproductive physiology. A knockdown of these genes caused lowered *vitellogenin* (*Vg*) expression in the fat body, reduced lipid accumulation in the oocytes and inhibition of patency in the follicular epithelium, thereby preventing oocyte growth [6,37]. A recent study in the desert locust, *Schistocerca gregaria*, confirmed these findings, since a knockdown of *Met* in female adult locusts resulted in a failure to initiate *Vg* expression in the fat body, as well as basal oocyte growth in the ovaries. Moreover, these ds*Met*-treated locusts showed delayed mating behavior and were unable to oviposit [38]. In female locusts, JH is known to stimulate Vg production by inducing *Vg* gene transcription in the fat body [39]. Via binding of JH to membrane receptor sites in the ovarian follicular epithelium, a process called patency, the shrinkage of cells, is initiated [40,41,42,43]. As a result, Vgs, which are mainly produced by the fat body and then released into the hemolymph, can be taken up by the developing oocyte. Moreover, an oviduct derived Patency Inducing Factor was recently discovered. This factor, clearly boosted by JH, is involved in the initiation of patency in the terminal follicle via the pedicel [44]. Apart from JH, ecdysteroids also play an important role in the reproductive physiology of female locusts. Produced in the ovarian follicular cell layer, they appear to induce meiotic re-initiation in the oocyte [45] and they are incorporated as conjugates into the eggs where they act as a source of ecdysteroids during embryogenesis [46]. Moreover, they can bind to vitellins, thereby preventing their leakage into the hemolymph [46,47,48]. Furthermore, in the desert locust *S. gregaria*, the ecdysteroid receptor complex is playing a crucial role in chorion formation [49].

Locust swarms are well known for their ability to destroy the agricultural production in some of the world’s poorest countries [50,51,52]. A major upsurge of *L. migratoria* in Madagascar reached biblical plague proportions in 2013 [53]. This plague endangered the food security of 13 million people and 2.3 million hectares of land had to be treated with insecticides in three consecutive control campaigns that lasted until July 2016. Moreover, several other locust species can form devastating swarms on different continents. Global warming together with poor monitoring due to political issues very recently led to a huge plague of desert locusts (*S. gregaria*) that are currently swarming through large parts of East Africa, the Middle East and the Indian subcontinent, threatening food supply and livelihoods of at least 20 million people [54]. Furthermore, in Sardinia, the recently occurring swarms of another species, the Moroccan locust (*Dociostaurus maroccanus*), have been described as the worst invasion in 60 years [55,56]. Unfortunately, the control of locust plagues mainly relies on the use of classic broad-spectrum insecticidal nerve toxins, which can result in a negative impact on the environment. Therefore, in the fight against locusts there is an urgent need for greener, more sustainable pest management strategies. Components of pathways regulating important insect processes, such as metamorphosis and reproduction, as investigated in the current study, would make for very attractive molecular targets of novel insect control agents. 

Our study describes a premature adult-like (adultoid) *L. migratoria* phenotype, the result of a reprogrammed development caused by the precocious RNAi-mediated silencing of *LmKr-h1* during the fourth (and normally penultimate) nymphal stage. Adultoid locusts are characterized by an adult color scheme and the presence of wings, although not fully extended, while maintaining the physical size of a fifth nymphal locust. Whereas several studies in different insect species already described the remarkable morphological consequences of such a knockdown, here we primarily focus on the reproductive properties of these adultoids and compare these with normal adult females. In this context, we analyze and compare the expression of genes contributing to the MEKRE93 axis, vitellogenesis and hormone biosynthetic pathways during the first gonadotrophic cycle of both adult and adultoid females. Moreover, effects on mating, oviposition, fecundity and fertility are monitored as well. 

## 2. Results

### 2.1. Temporal Transcript Profiles of MEKRE93 and JH Biosynthesis Pathway Components

In female locusts (double-stranded green fluorescent protein (*dsGFP*)-injected control), we investigated the temporal profiles of the MEKRE93 pathway transcripts *LmKr-h1*, *LmE93* and *LmMet*, as well as these of *LmJHAMT* and *LmCYP15A1*, the enzymes catalyzing the two final steps in JH biosynthesis, by qRT-PCR analyses at different time points from day 4 of the fourth nymphal (N4 D4) until day 12 of the adult (Ad D12) stage (Appendix A). Relative *LmKr-h1* transcript levels were studied in the fat body, ovaries and CA. In the fat body, *LmKr-h1* was highly expressed at the end of the fourth nymphal stage (N4 D4). After molting to fifth nymphal locusts (N5 D0), *LmKr-h1* levels were significantly lower and stayed low at the following time point until initiating the adult molt (N5 D6 and Ad D0) when significant increases were observed (Appendix A). A gradual increase in *LmKr-h1* expression was observed in the ovaries of adult females (Appendix A), whereas in the CA (Appendix A), reduced *LmKr-h1* levels were detected on the days of molting (N5 D0, Ad D0) and when the ovaries had entered the vitellogenic stage (Ad D12). At the two earliest time points, situated before and after the N4-N5 molt (N4 D4 and N5 D0), *LmE93* levels were low in both fat body and CA, but rose significantly during the final nymphal stage (N5 D3) (Appendix A). In fat body, *LmE93* expression remained high at the different time points in the adult stage, while in the CA of the adult stage they gradually decreased towards day 12 after the final molt (Ad D12). In addition, *LmMet* expression levels in fat body were high on day 4 of the fourth nymphal stage (N4 D4) followed by a drop towards the fifth nymphal stage (N5 D0), after which they remained relatively constant at the following time points in this nymphal stage. In fat body of adult females, significantly lower *LmMe*t levels were detected on days 4 and 12 versus higher ones on days 0 and 8 (Appendix A). *LmMet* expression levels measured in the CA showed a decreasing trend throughout the different time points of the final instar and adult stage. For checking the expression of two JH biosynthetic enzymes, the site of JH biosynthesis, i.e., the CA, was selected. Both *LmJHAMT* and *LmCYP15A1* showed a significant increase in their expression after the transition from fifth nymphal to adult locusts (Ad D0), and *LmJHAMT* further increased on day four of the adult stage (Ad D4). The relative expression levels of both genes remained high at the time points observed during adult life (Appendix A).

### 2.2. Knockdown of LmKr-h1 Results in Precocious Development of External Adult Features

An initial RNAi pilot study was performed to eliminate the possibility of off-target effects and to verify the reproducibility of the phenotypic effects. Therefore, two distinct dsRNA constructs were designed (*dsLmKr-h1* and *dsLmKr-h1_2;*
Appendix A). When tested within the same experimental setup, similar phenotypic effects were observed with both constructs. Next, the *dsLmKr-h1* construct was utilized in all further experiments. 

The RNAi treatment, based on injections of *dsLmKr-h1* during the fourth nymphal stage (on D0 and D3), significantly affected development of *L. migratoria* (Figure 1). In total, 93% of the experimental locusts did not molt into the fifth nymphal stage, while 100% of the *dsGFP*-injected (control) locusts successfully completed this nymphal-to-nymphal molt (Figure 1A–C). Instead of molting into fifth nymphal locusts, most of the *dsLmKr-h1*-injected (experimental) animals molted into an adultoid phenotype, showing precocious development of adult features. These adultoids already had an adult color scheme as well as the presence of wings, although not fully extended and non-functional, while maintaining the physical body size of a fifth nymphal locust (Figure 1A). The animals that molted to the adultoid phenotype had remained in the fourth nymphal stage for a longer time than the *dsGFP*-injected (control) locusts that molted to the fifth nymphal stage: their N4 stage lasted for 6–8 days compared to 4–6 days, respectively (Figure 1C). Moreover, the adultoid insects did not undergo any further molts (Figure 1D). Apart from the adultoid phenotype, 29% of the *dsLmKr-h1-injected* (experimental) animals molted into an intermediate phenotype (Figure 1B), still having a nymphal color scheme, but with more developed wings (Figure 1A). These animals had remained in the fourth nymphal stage for 5 to 7 days before molting to their intermediate phenotype (Figure 1C). Unlike the adultoid phenotype, 20% of the intermediate phenotype locusts were still successfully completing the adult molt, while 100% of the control (*dsGFP*) and experimental (*dsLmKr-h1*) fifth nymphal locusts successfully completed the adult molt (Figure 1D).

### 2.3. Transcript Levels of MEKRE93 and JH Biosynthesis Pathway Components in dsRNA-Injected Fourth Nymphal and Freshly Molted Fifth Nymphal, Adultoid and Adult Female Locusts

Transcript levels of different components of the MEKRE93 pathway, as well as the enzymes catalyzing the two final steps of JH biosynthesis, were determined in different tissues on two developmental time points, i.e., day 4 of the fourth nymphal stage (N4 D4) and freshly molted fifth nymphal (N5 D0), adult (Ad D0) or adultoid (Ao D0) locusts (Appendix A). A significant downregulation of *LmKr-h1* transcript levels was measured in the fat body (44%) of fourth nymphal *dsLmKr-h1*-injected locusts, when compared with *dsGFP*-injected (control) locusts of the same age (N4 D4; Appendix A). While in the ovaries (Appendix A) no significant change in the transcript level was observed at this time point, a significant increase was noticed in the CA (Appendix A). Interestingly, on the day of molting, *LmKr-h1* relative transcript levels were significantly higher in fat body and ovaries of adultoid locusts than fifth nymphal ones (Appendix A), while no significant differences were observed in the CA (Appendix A). 

In comparison with *dsGFP*-injected (control) locusts, the *LmE93* mRNA levels were significantly higher in both fat body and CA of *dsLmKr-h1*-injected locusts on day 4 of the fourth instar (N4 D4). On the day of molting, the *LmE93* transcript levels in both tissues were significantly higher in adultoid (*dsLmKr-h1*) than in fifth nymphal locusts (*dsGFP* control), while no significant differences were observed between adultoid and adult locusts (Appendix A). For *LmMet* mRNA levels, on the other hand, no significant changes were noticed with the single exception of a significantly lower level in the fat body of freshly molted fifth nymphal (*dsGFP* control) locusts (Appendix A). When checking the expression levels of two JH biosynthetic enzymes, *LmJHAMT* and *LmCYP15A1* in the CA, no significant differences were observed between *dsLmKr-h1* and *dsGFP*-injected locusts on the time points that were analyzed (Appendix A). 

### 2.4. The Adultoid Phenotype Shows an Accelerated Ovarian Maturation

To investigate the effect of the RNAi-mediated knockdown of *LmKr-h1* in fourth nymphal locusts on the later female reproductive physiology, adult (*dsGFP* control) and adultoid (*dsLmKr-h1*) locusts were dissected every four days (Figure 2A). Adultoid female locusts were in full vitellogenesis eight days after their precociously induced final molt (Ao D8). In comparison, eight days after their final molt (Ad D8) the adult females included in the same experiment had an average oocyte length of only 1.39 mm. Moreover, considering that a fifth nymphal stage preceded their adult molt, as illustrated in Figure 2A, the locusts of the control group (*dsGFP*) were, in total, at least four days older than the adultoid (*dsLmKr-h1*) females. Adultoid females did not only develop faster but their ovaries also matured faster, as shown by the growth of their basal oocytes over time (Figure 2B). On day 4 of the adult stage, the average basal oocyte length measured 0.62 mm, while in adultoids this was 0.53 mm. Significantly accelerated growth was observed on day 8 of the adultoid phenotype with the average oocyte length of 3.88 mm, more than double the size of the basal oocytes in day 8 adults measuring 1.39 mm. A significant length difference was also observed between the basal oocytes in adultoids and adults on day 12 after the final molt. Ovaries were further dissected, cleaned by removing the attached fat body and imaged (Figure 2C–E’ and M–O). Some adultoid females were already in their second gonadotrophic cycle, characterized by orange lesions at the base of the new basal oocyte. In addition to these orange lesions, multiple resorbing oocytes were observed (Figure 2D–E’). Upon dissection of the adultoid females, it became clear that, while most of these animals were unable to deposit their eggs, fully grown oocytes were filling the oviducts (Figure 2E). Transverse sections of terminal oocytes were examined. The sections clearly confirmed the presence of yolk (blue matrix) and lipid droplets (grey spheres) from as early as eight days after the final molt to adultoid (Figure 2F–H), whereas in the oocytes of adult (control) locusts, the presence of yolk and lipid droplets was only observed at day 12 (Figure 2K,L). In the resorbing oocytes of adultoids, a loss of yolk was observed (Figure 2I,I’). The oocytes found in the oviducts of adultoid locusts contained yolk and lipid droplets and were clearly surrounded by a chorion (Figure 2J). 

Since yolk accumulation was observed relatively early in oocytes of adultoid (*dsLmKr-h1*) females, *LmVg* mRNA levels were analyzed in fat body samples during the first gonadotrophic cycle and compared to normal (*dsGFP* control) adult females (Figure 3). In the fat bodies of adultoid locusts, significantly higher *Vg* mRNA levels were observed for both *LmVg1* (Figure 3A) and *LmVg2* (Figure 3B) on days 8 and 12 after the final molt. When checking the temporal profiles of *LmVg* mRNA transcripts in fat body samples of adult and adultoid female locusts during their first gonadotrophic cycle (Appendix A), significantly higher *LmVg* mRNA levels were observed for both *LmVg1* (Appendix A) and *LmVg2* (Appendix A) on day 8 compared to day 4. Unlike adultoid females, *LmVg* mRNA levels in control adult females were still significantly increasing towards day 12 (Appendix A). 

### 2.5. Transcript Levels of MEKRE93 and JH Biosynthesis Pathway Components in Adultoid Females

The temporal expression profiles of genes encoding MEKRE93 and JH biosynthesis pathway components were monitored in the fat body, in the CA and in the ovaries of adultoid (*dsLmKr-h1*) female locusts (Appendix A). Figure 4 compares the expression levels at different time points after the final molt between adultoid (*dsLmKr-h1*) and adult (*dsGFP* control) female locusts. In the fat body of adultoid female locusts, *LmKr-h1* transcript levels significantly increased at day 8 (Appendix A). At this time point a very pronounced increase was also observed in the ovaries (Appendix A), whereas in the CA an increasing trend was observed from D4 to D12 (Appendix A). When compared to adult females, the relative *LmKr-h1* mRNA levels of adultoid female locusts were significantly higher at different time points after the final molt: on days 4 and 12 in the fat body and day 12 in the CA (Figure 4A,B). In the fat body of adultoids, *LmE93* and *LmMet* mRNA levels were significantly higher on the day of molting (Ao D0) and then decreased on the following time point (Ao D4) (Appendix A). Furthermore, when compared to days 0 and 4, a very pronounced decrease in the *LmE93* level was observed in the fat body of adultoids on day 8 (Ao D8) (Appendix A). In the fat body of adult female locusts, a clear decreasing trend in the *LmE93* and *LmMet* expression profiles was not observed (Appendix A). Therefore, in comparison to the adults, the adultoids had significantly lower *LmE93* and *LmMet* transcript levels in their fat body on day 8 after the final molt (Figure 4C,E). On the other hand, when compared to adult females, the adultoids had significantly higher *LmE93* transcript levels in the CA at some time points (D4 and D12), while for *LmMet* no significant differences were observed between adults and adultoids in these JH producing endocrine organs (Figure 4D,F). In a Pearson correlation calculation, the temporal profiles of the relative mRNA levels of *LmE93* and *LmMet* in the CA of control (*dsGFP*) adult females correlated well with these of adultoid (*dsLmKr-h1*) females (Figure 4D,F). For the two JH biosynthetic enzymes, *LmJHAMT* and *LmCYP15A1*, relative transcript levels tended to be higher in sexually maturing adultoid females (Appendix A). When compared to adults, the adultoid females had significantly higher *LmCYP15A1* levels in their CA on day 8 after the final molt (Figure 4H). 

### 2.6. Transcript Levels of Halloween Genes and Ecdysone Receptor Complex in Adultoids And Adults

Ovaries were collected every four days starting from freshly molted (D0) adultoid and adult locusts until day 12 (D12) after the final molt. Figure 5 compares the ovarian expression levels of *LmKr-h1*, the *Halloween* genes, *Spook* (*LmSpo*), *Phantom* (*LmPhm*), *Disembodied* (*LmDib*), *Shadow* (*LmSad*) and *Shade* (*LmShd*), which code for cytochrome P450 enzymes involved in ecdysteroid biosynthesis, as well as of the *LmEcR* and *LmRXR* genes, which encode the components of the ecdysone receptor complex, in adultoid (*dsLmKr-h1*) and adult (*dsGFP* control) female locusts at these different time points. Appendix A show the temporal expression profiles for *Halloween* genes and ecdysone receptor components in ovaries of adult (*dsGFP* control) and adultoid (*dsLmKr-h1*) females, respectively. 

In adult females, a significantly increased expression level was observed on day 12 (Ad D12) for *LmPhm*, while a similar temporal trend was also noticeable for *LmSpo* and *LmSad* (Appendix A). For *LmShd* and *LmEcR* no significant changes in expression were observed (Appendix A), while *LmDib* and *LmRXR* levels seemed to progressively decline at different time points after the final molt (Appendix A). No correlations were found between adult and adultoid locusts for the temporal profiles of ovarian *LmKr-h1*, *LmSpo, LmPhm*, *LmDib*, *LmSad*, *LmShd*, *LmEcR* and *LmRXR* transcripts. 

In comparison with freshly molted adultoid females (Ao D0), *LmPhm*, *LmEcR* and *LmRXR* transcript levels were significantly reduced on day 4 (Ao D4) after the final molt (Appendix A). Furthermore, in these adultoids the *LmSpo, LmPhm, LmSad* and *LmEcR* genes showed significantly increased transcript levels on day 8 (Ao D8) (Appendix A). At this time point, the rise of *LmSpo* and *LmPhm* levels in adultoids appeared very pronounced, as on day 8 (D8) the transcript levels for both *Halloween* genes were also significantly higher than in adult females (Figure 5B,C). A similar temporal profile was noticed when investigating *LmKr-h1* expression levels in the ovaries of adultoid locusts (Appendix A and Figure 5A). For *LmDib* no significant changes in expression were observed between the different time points analyzed in the adultoids (Appendix A). Nevertheless, in comparison with adult (control) females, a significantly higher *LmDib* mRNA level was observed on day 12 (D12) (Figure 5D). In addition, when compared to control adults, significantly lower *LmSpo, LmSad* and *LmRXR* transcript levels were detected in the adultoids on day 4 (D4) (Figure 5B,E,H). No significant differences in *LmShd* and *LmEcR* expression were observed between adult and adultoid females at the different time points (Figure 5F,G). 

### 2.7. Adultoid Females Have Severe Defects in Mating and Egg Laying 

Mating, fecundity and fertility were assessed in the adultoid (*dsLmKr-h1*) locusts in comparison with normal adults (*dsGFP* control). Therefore, the events of mating with a virgin adult male (showing an actual connection between the male and female genitals) and egg deposition were monitored for adult and adultoid females (Figure 6). We also registered the period between mating and oviposition, the incubation time from egg laying until hatching, as well as the numbers of eggs and hatchlings per egg pod that had been deposited by the females (Appendix A). Our results indicate that the mating process was significantly affected in the adultoid females. Adultoid females did not mate with virgin control males within the experimental timeframe (Figure 6A). In comparison, all control females mated successfully (Figure 6A). When paired with adultoid males, only a small fraction (14%) of the adultoid females was mating (Figure 6A and Appendix A). Oviposition was also severely impaired, since only the adultoid females that had mated also deposited their eggs (Figure 6B and Appendix A). Moreover, they deposited their eggs significantly earlier than the adult females (Appendix A). Interestingly, the eggs deposited by the adultoid females appeared similar in size and shape to the ones obtained from adult females. However, the incubation period between egg laying and hatching was significantly longer (Appendix A). Appendix A shows the numbers of eggs and hatchlings per egg pod. Although adult and adultoid females did not significantly differ in the number of eggs they deposited per pod (fecundity), the number of hatchlings (fertility) was significantly lower for the offspring of adultoid locusts.

## 3. Discussion

### 3.1. DsLmKr-h1 Injections in Fourth Nymphal Stage Induced the Developmental Transition to Adultoids

The migratory locust, *L. migratoria,* is a hemimetabolan insect that normally passes five nymphal stages before the adult, reproductive stage is reached. The final molt is characterized by an incomplete metamorphosis process, which results in development of functional wings and reproductive organs. In this study, a final molt was precociously induced by RNAi-mediated knockdown of *LmKr-h1*, a well-conserved transcription factor acting immediately downstream of the JH receptor Met, during the fourth nymphal stage (Figure 1). The adultoid phenotype resulting from this treatment displayed the coloration pattern that is typical for adults. On the other hand, its wings were not fully extended. Similar observations of precocious adult-like phenotypes were made after *Kr-h1* knockdown in several other hemimetabolan insect species, such as *B. germanica*, *R. prolixus*, *G. bimaculatus*, *N. lugens*, *C. lectularius* and *P. apterus* [10,11,13,14,15,16,17,27]. 

Locusts were injected with *dsLmKr-h1* on the day of their molt to the fourth instar and a second injection was given 3 days later. One day later (N4 D4), a significant downregulation of the *LmKr-h1* transcript level was detected in the fat body (Appendix A). On the day of the adultoid molt (Ao D0), the *LmKr-h1* levels were clearly much higher than in the *dsGFP*-injected (control) N5 condition and resembled the normal adult levels (Ad D0) (Appendix A), suggesting that the effect of the knockdown was temporally restricted to the N4 stage. Similar studies in the common bed bug, *C. lectularius,* and the German cockroach, *B. germanica,* showed similar efficiencies of *Kr-h1* transcript knockdown [14,17]. The majority (64%) of the *dsLmKr-h1*-injected locusts developed into adultoids (Figure 1A and B). The presence of some fifth instar and intermediate phenotypes, as also observed by Lozano and Belles (2011) in *B. germanica* upon silencing *Kr-h1*, suggests some inter-individual variation in knockdown efficiency, which may account for the relatively mild reduction in the *LmKr-h1* transcript level that was measured in *dsLmKr-h1*-injected locusts on day 4 in the fourth nymphal stage [14]. A second injection in *B. germanica* was sufficient to obtain 100% adultoids, but this was not the case in the locusts of this study. The difficulty of obtaining adultoids with full efficiency (100%) was also noticed in the common bed bug, *C. lectularius.* When knocking down *Kr-h1* in the penultimate instar, only 31% of the dsRNA-injected insects developed precociously into adultoids [17]. It is possible that the spatiotemporal characteristics of the RNAi-mediated knockdown influence its developmental outcome. In our study no downregulation of *LmKr-h1* was observed in the ovaries (Appendix A), which can be explained by the previously described inefficient uptake of dsRNA into the follicle cells and oocytes [57,58]. Remarkably, a significantly increased *LmKr-h1* transcript level was detected in the CA (Appendix A). This suggests the existence of a very dynamic and tissue-dependent regulation of this immediate early JH response gene, which may explain why its downregulation was not maintained longer and even resulted in a compensatory effect within the CA. Still, the phenotypic effects obtained in this experiment were very obvious and fully in line with the well-established role of Kr-h1 in controlling the developmental timing of metamorphosis. 

Expression of *Kr-h1* is known to be controlled by JH activity, downstream of Met, as previously reported for several insect species [8,10,12,14,59]. Our qRT-PCR measurements indeed show that in *dsGFP*-injected control locusts, the expression of both *LmKr-h1* and *LmMet* in fat body was reduced on the day of molting to the fifth nymphal stage (N5 D0), followed by increased transcript levels of the adult specifier *Lm*E93 on the next time point that was analyzed (N5 D3) (Appendix A). These temporal profiles are in agreement with the described role of these factors in the control of metamorphosis. Interestingly, for the *dsLmKr-h1*-injected locusts that developed into adultoids, the total duration of the fourth nymphal stage (from N4 D0 until the N4-adultoid transition) was on average two days longer than that of a normal fourth nymphal stage (5 days) and thus similar to that of a normal fifth nymphal stage (7 days) (Figure 1C). Moreover, none of the adultoids and only a small fraction of the intermediate phenotypes resulting from the *LmKr-h1* knockdown had any further molts (Figure 1D). These observations indicate that the transition process from N4-nymph to adultoid can indeed be considered as a precocious metamorphosis, at which the locusts were reprogrammed to stop increasing their body size and start investing more energy and nutrients in reproduction. Furthermore, clear differences in transcript levels of MEKRE93 pathway components were observed during this time. Normally, the expression levels of the adult specifier gene *E93* only start rising in the final nymphal stage, when JH titers drop and ecdysteroid titers rise, to ensure the final transition to the adult stage [21]. A significant rise in *E93* levels was indeed observed on day three (N5 D3) in the control fifth nymphal locusts (Appendix A). Therefore, the significantly higher *LmE93* levels that were detected in the *dsLmKr-h1*-injected locusts on day 4 of the fourth nymphal stage (Appendix A) can explain the transition of the majority of these locusts into adultoids (Figure 1B). Ishimaru et al. (2019) also reported elevated *E93* transcript levels in premature *G. bimaculatus* adults after knocking down *Kr-h1* [15]. A similar observation was made by Lozano and Belles (2011) who described that female fifth instar *B. germanica,* treated with *dsKr-h1*, required on average two to three days more to perform the next molt [14]. *LmMet* expression measured in the fat body remained relatively stable on different time points during the fifth nymphal stage of control animals (Appendix A). Similar temporal profiles were observed in *B. germanica*, *N. lugens* and *P. apterus*, with *Met* transcript levels staying relatively stable in nymphal whole body RNA samples, independently of JH titers [10,11,12,22]. However, in the CA of locusts, the *LmMet* levels appeared to gradually decline over time (Appendix A). Interestingly, in freshly molted adultoids the transcript levels of MEKRE93 pathway components were more similar to the ones observed in adults than in fifth instar control locusts on the day of molting (Appendix A), emphasizing that the adult-like properties of the adultoid locusts were also noticeable at the molecular level.

### 3.2. Comparison Between Adultoid and Adult Female Locusts during Their First Gonadotrophic Cycle

In the current study, we mainly focus on the reproductive physiology of the adultoid females and compare this with normal adult female locusts (*dsGFP*-injected control). In this context we also analyzed the transcript levels of several genes involved in hormone synthesis, signal transduction and vitellogenesis. Our data indicate that the adultoid females have the following characteristics: 

#### 3.2.1. An Accelerated Ovarian Maturation

The adultoid females were characterized by an accelerated entry into the vitellogenic stage of the first gonadotrophic cycle (Figure 2). The first gonadotrophic cycle in the ovaries of female locusts can be divided in five stages: (i) an early growth period; (ii) the previtellogenic stage; (iii) the vitellogenic stage; (iv) the choriogenic stage and (v) ovulation and oviposition [60,61]. It is clear from Figure 2 that the adultoid females entered the vitellogenic stage several days prior to the adult control locusts. This stage is characterized by patency of the follicular epithelium, allowing the passage of lipids and vitellogenins from the hemolymph to the developing basal oocytes, causing them to grow. Although the basal oocytes in adultoid ovaries were slightly smaller on day four after the final molt, they became much larger than these in adults on the subsequent time points (D8 and D12 in Figure 2B; Figure 2D versus 2N). To further confirm this early incorporation of yolk material, transverse sections of the oocytes were made (Figure 2F–L). Whereas in oocytes of adultoid locusts, proteins (blue matrix) and lipid droplets (greyish droplets) were already visible on day 8 after the final molt (Figure 2G), it took oocytes of adult locusts approximately four additional days to enter this stage (Figure 2L). In addition, some adultoid ovaries had then already entered the second gonadotrophic cycle and also contained resorbing oocytes (Figure 2D–E’, I–I’). When the eggshell or chorion was formed, the basal oocytes took on a shiny appearance, then ovulation occurred and fully grown oocytes entered the lateral oviducts (Figure 2E). Morphological analysis of these oocytes confirmed the presence of a chorion (Figure 2J). 

#### 3.2.2. Associated Changes in Gene Expression Profiles in Fat Body and CA 

Vitellogenin synthesis was monitored at the mRNA level on different time points in both adult and adultoid female locusts. When compared to adult females, the fat body of adultoids contained significantly higher levels of both *LmVg1* and *LmVg2* on days eight and twelve after the final molt (D8 and D12 in Figure 3). This timing is fully in line with the observed increase in length of the basal oocytes occurring in the adultoid female ovaries (Figure 2B). In addition, significantly higher transcript levels of the immediate early JH response gene *LmKr-h*1 were also detected in fat body of adultoid females compared to control adult females (Figure 4A). The importance of JH signaling in regulating female vitellogenesis was previously confirmed in the migratory locust, *L. migratoria*. A study by Song et al. (2014) described reduced *Vg* gene expression in the fat body, lipid accumulation in the primary oocytes, as well as a strong inhibition of oocyte maturation upon silencing *LmMet* and *LmKr-h1* [6]. These observations are corroborated by multiple other studies performed in numerous insect species [13,23,24,25,27,29,31,32,33,34,38,62,63,64]. In addition to the higher *LmKr-h1* transcript levels detected in fat body of adultoid females (Appendix A), lower *LmE93* transcript levels were observed (Appendix A), which may refer to the inhibitory effect of Kr-h1 on *E93* expression [21,65]. *LmE93* expression was extremely low on day 8, when the adultoid females were in full vitellogenesis (D8 in Figure 4B). However, in fat body of normal adult females an inverse relationship between *LmKr-h1* and *LmE93* gene expression was not noticeable (Ad D0-D12 in Appendix A), which may point at an important difference upstream in the regulation of these genes between adults and adultoids. In the CA, *LmJHAMT* and *LmCYP15A1* expression levels were increased in the adult/adultoid stages, which may refer to the important roles that are played by JH in reproduction (Appendix A). Moreover, the relative expression level of *LmCYP15A1* was higher on day 8 (Figure 4H) in the vitellogenic adultoid females, when compared with the control adult females. Multiple studies also reported on the increased expression of JH biosynthetic enzymes during the gonadotrophic cycle of two different locust species [6,38], as well as several other insects [29,66,67,68,69,70,71]. During the maturation of reproductive organs and progression of the first gonadotrophic cycle, JH signaling is also crucial for the induction of polyploidy in the fat body cells, allowing for the massive metabolic conversion of stored energy and nutrients into yolk constituents, such as vitellogenins and lipids, which eventually will be incorporated into the basal oocytes during the vitellogenic phase [63,64].

#### 3.2.3. Accelerated *Kr-h1* and *Halloween* Gene Expression in the Ovary 

In addition to JH, ecdysteroids are playing an important role in the reproductive physiology of female locusts. Ecdysteroid biosynthesis mainly takes place in the follicular cell layer of the developing oocytes [72,73]. In a series of enzymatic reactions involving the *Halloween* genes, the steroid hormone precursor cholesterol is converted to 20E [74]. When comparing female adultoids with control adults, significant differences in *Halloween* gene expression were observed on different time points after the final molt (Figure 5). In female adultoids, day 8 appeared to be a critical time point when a prominent peak of both *LmSpo* and *LmPhm* expression levels was observed (Appendix A). Where *Spo* is considered to catalyze the rate-limiting step, *Phm* is important in the conversion of ketodiol to ketotriol [75,76,77]. In agreement with the current study (Appendix A), Marchal et al. (2011) also showed that the transcript levels of both *Halloween* genes were low in freshly molted adults and rose during the female reproductive cycle of the desert locust, *S. gregaria* [71]. The observed elevated *LmSpo* and *LmPhm* transcript levels in the ovaries of adultoids on day 8 after their final molt, coinciding with the significantly increased *Kr-h1* levels in the same organs (Appendix A), therefore once again reinforce our observations of an accelerated ovarian maturation in comparison with normal adult migratory locusts (D8 in Figure 5B,C). In addition to the accumulation of ecdysteroid conjugates in basal oocytes during vitellogenesis, acting as a source of ecdysteroids during embryonic development, ecdysteroid signaling also plays a role in choriogenesis, involving the heterodimeric nuclear receptor complex consisting of the ecdysone receptor (EcR) and the retinoid-X-receptor (RXR). An RNAi-mediated knockdown of *EcR/RXR* in the desert locust, *S. gregaria,* and the Pacific beetle cockroach, *D. punctata,* resulted in females incapable of initiating choriogenesis; basal oocytes reached their maximal length, after which they were resorbed, thereby affecting ovulation as well [49,78]. These studies also suggested that EcR and RXR mediate critical ecdysteroid feedback to the CA, essential at the end of vitellogenesis for successful ovulation. In 1993, Belles et al. already suggested the existence of a crosstalk between JH and ecdysteroids in *B. germanica,* since ecdysteroids are responsible for the termination of the gonadotrophic cycle via lowering JH synthesis [79]. In our study, the ovarian *LmEcR* and *LmRXR* transcript levels did not differ significantly on the observed time points during the vitellogenic stage between adult and adultoid female locusts (D8 and D12 in Figure 5G,H). Nevertheless, in adultoid ovaries, the *LmEcR* levels significantly increased at day eight, which may refer to the crucial role of ecdysteroid signaling in choriogenesis (Ao D8 in Appendix A). These results can explain the normal appearance of the fully grown oocytes observed in adultoid females (Figure 2J).

#### 3.2.4. Severe Defects in Mating and Egg Laying 

None of the adultoid females effectively mated with virgin control males (Figure 6A). Since differences in body size and anatomy between adultoid and adult locusts (Figure 1A and Appendix A) could have an adverse effect on mating success, adultoid females were also paired with adultoid males (Appendix A). In addition, in this setup, mating was negatively affected when compared to control adult locusts (Figure 6A). To verify whether adultoid males were capable of mating, some were also paired with control females, eventually confirming that mating was physically possible (Appendix A). The few adultoid females that were able to mate needed fewer days to deposit their eggs (Figure 6B and Appendix A). However, it took the few hatchlings that were produced (7.5%) several days longer to emerge (Appendix A). Remarkably, all adultoid females, even if they did not mate, attempted to oviposit, but the unmated ones did not succeed and subsequently died in the coming days. Dissection of their abdomens revealed ovaries full of mature oocytes, some of them stuck in the oviducts and leaking yolk (Appendix A). In *L. migratoria*, mating is known to induce physiological and endocrinological changes in the female resulting in appropriate maturation of the eggs and the subsequent deposition of fertile eggs in a suitable environment [80]. Mating in the kissing bug, *R. prolixus,* triggers the release of a myotropic peptide from the nervous system, resulting in ovulation and oviposition [81]. In *Drosophila melanogaster*, the Sex Peptide is present in the seminal fluid and transferred from male to female upon mating; in the female fly, it triggers a recently described neural circuitry that controls egg laying [82]. Therefore, the incapability of virgin adultoid *L. migratoria* females to oviposit might be due to the absence of such a physiological stimulus induced by mating. In addition, a recent study in the cricket, *G. bimaculatus,* where silencing *Kr-h1* resulted in adultoid females, described the occurrence of an abnormal ovipositor morphology [15]. Although the ovipositors of the adultoid female locusts had a normal anatomical appearance, their size was smaller when compared to the adult ones (Appendix A). Since the eggs of adultoids had a similar size as the ones produced by adults, it is not unlikely that this may have caused great discomfort during oviposition. Finally, while the number of eggs per egg pod (fecundity) produced by the adultoid females was similar to that of adults, the number of hatchlings (fertility) was severely reduced. The reasons for this low fertility are not fully clear. We observed that adultoid males were eager to mate with an adultoid or adult female and noticed that their testes and accessory glands had a normal adult-like appearance (Appendix A). However, we do not know how much sperm was transferred upon mating and how fertile it was. Therefore, the observed hatchlings either developed from fertilized eggs or were generated by parthenogenesis [83,84]. Future dissections of adultoid male testes and adultoid female spermatheca, to verify the presence and transfer of sperm, would probably enable a more comprehensive interpretation of these data. Nevertheless, the fact that some eggs successfully hatched indicates that some adultoids were capable of generating offspring.

## 4. Materials and Methods

### 4.1. Rearing of Animals

The migratory locusts (*L. migratoria*) were reared under crowded conditions in cages that were situated in a room with controlled temperature (32 ± 1 °C) and day/night cycle (photophase of 13 h) at an ambient relative humidity between 40% and 60%. The animals were fed freshly sprouted wheat leaves and dry oat flakes ad libitum. Pots containing a slightly moistened mixture of sand and turf were placed in the breeding cages allowing mated females to deposit their eggs. These pots were collected once a week and transferred to clean cages where eggs were allowed to hatch into first instar locusts.

### 4.2. Tissue Collection

The tissues of interest were dissected under a binocular microscope in *L. migratoria* Ringer solution (1 L: 9.82 g NaCl; 0.32 g CaCl_2_; 0.48 g KCl; 0.73 g MgCl_2_; 0.25 g NaHCO_3_; 0.19 g NaH_2_PO_4_; pH 6.5) and snap-frozen by pooling them in MagNA Lyser Green Beads Tubes (Roche, Mannheim, Germany) or RNase-free Screw Cap Microcentrifuge tubes which were placed in liquid nitrogen to prevent RNA degradation. Tissues were collected in four pools of three animals each and were used in an RNA extraction protocol or stored at −80 °C until further processing. 

### 4.3. RNA Extraction and cDNA Synthesis

Fat body and ovary samples of female migratory locusts were pooled in MagNA Lyser Green Beads Tubes (Roche) and homogenized using a MagNA Lyser instrument (30 sec, 5000 rpm; Roche). Total RNA was extracted using the RNeasy Lipid Tissue Kit (Qiagen, Austin, TX, USA) according to the manufacturer’s protocol. An on-column DNase digestion (RNase-free DNase set, Qiagen) was performed to prevent genomic DNA contamination. The CA were pooled in RNase-free Screw Cap Microcentrifuge tubes. According to the manufacturer’s protocol, total RNA was extracted using the RNAqueous-Micro Kit (Ambion; Life Technologies, Carlsbad, CA, USA), a specialized kit for total RNA extraction of small sized tissues. A DNase incubation step was included here as well. Subsequently, the quality and concentration of the resulting RNA samples were measured using a Nanodrop spectrophotometer (NanoPhotometer N60, Implen, München, Germany). From each RNA sample equal amounts (150 ng for fat body and ovaries and 50 ng for CA) were reverse-transcribed using a mix of random hexamers and oligo(dT) primers according to the manufacturer’s protocol (PrimeScript^TM^ RT Reagent Kit, TaKaRa, Invitrogen Life Technologies, Carlsbad, CA, USA). The obtained cDNA was then diluted ten-fold with Milli-Q water (Merck Millipore, Darmstadt, Germany).

### 4.4. Quantitative Real-Time PCR

QRT-PCR primers for the different target genes (Appendix A) were designed using Primer Express software (Applied Biosystems, Foster City, CA, USA). These primer pairs were validated by designing relative standard curves for gene transcripts with serial (5x) dilutions of appropriate cDNA samples. The correlation coefficient and efficiency of the qRT-PCR reaction were measured (R^2^ = 0.995–1; Eff% = 90%–110%) for each primer pair. All qRT-PCR reactions were performed in duplicate in 96-well plates on a StepOne Plus System (ABI Prism, Applied Biosystems, Foster City, CA, USA) according to the Fast SYBR Green PCR Master Mix protocol. Each reaction contained 5.0 µL of fast SYBR Green, 0.5 µL of each Forward and Reverse primer (10 µM), 1.5 µL of Milli-Q water and 2.5 µL of cDNA. “No template control” reactions were included to confirm absence of contamination; RT^-^ reactions were performed to check for possible genomic DNA contamination. The following thermal cycling profile was used: 95 °C for 10 min, followed by 40 cycles of 95 °C for 15 s and 60 °C for 60 s. Afterwards, a melt curve analysis was performed to confirm the specificity of the qRT-PCR reactions. Additionally, the amplification products were analyzed using horizontal agarose gel electrophoresis (1.2% agarose gel containing GelRed^™^, Biotium, Fremont, CA, USA) and visualized under UV light. Only a single band of the expected size for each transcript was observed, further cloned and sequenced (TOPO^®^ TA Cloning Kit for sequencing, Invitrogen, Carlsbad, CA, USA) to confirm target specificity. 

Optimal housekeeping genes were selected using geNorm software [85]. In the designed experiment *rps13* + *CG13220*, *CG13220* + *TubA1* and *rp49* + *rps13* appeared to be most stable in the studied fat body, ovary and CA samples, respectively. Relative expression levels were determined according to the comparative Ct method (ΔΔCt) [85]. QRT-PCR was used to determine the temporal distributions of the transcripts of interest. Moreover, statistically significant differences between the *dsLmKr-h1* and *dsGFP* (control)-injected female locusts were found via a *t*-test on the log transformed data (with or without two-sided Welch’s correction) using GraphPad Prism 6 (GraphPad Software Inc., San Diego, CA, USA). The correlation coefficient (r) between the transcript profiles of adult and adultoid locusts was obtained by a Pearson correlation calculation. 

### 4.5. RNA Interference Experiments

Double-stranded (ds)RNA constructs for *LmKr-h1* and *GFP* (control) were produced using Ambion’s MEGAScript RNAi kit following the manufacturer’s protocol (Appendix A). This procedure is based on a high-yield in vitro transcription reaction from a user-provided DNA template with a T7 promoter sequence at the 5′ ends of each strand. Forward and reverse primers flanked by the T7 promoter sequence (given in Appendix A) were used in a PCR reaction with REDTaq DNA polymerase (Sigma-Aldrich, Darmstadt, Germany) to amplify a fragment of the target gene. The resulting amplicons were analyzed using horizontal agarose gel electrophoresis (1.2% agarose gel containing GelRed^TM^, Biotium, Fremont, CA, USA) and visualized under UV light. Only a single band was observed, which was subsequently cloned and sequenced (TOPO^®^ TA Cloning Kit for sequencing, Invitrogen, Carlsbad, CA, USA) to confirm target specificity. The verified constructs were used as template in a high-yield in vitro transcription reaction with T7 RNA polymerase. Remaining ssRNA and DNA were removed in a nuclease digestion step. Quality and concentration of the produced dsRNA were determined using a Nanodrop spectrophotometer (NanoPhotometer N60, Implen, München, Germany). To confirm dsRNA integrity, a small amount of the reaction product was checked by horizontal agarose gel electrophoresis (1.2% agarose gel containing GelRed^TM^, Biotium, Fremont, CA, USA). 

83 Newly molted fourth nymphal (N4) female locusts, synchronized on the day of ecdysis (D0), were injected with 6 µL dsRNA against *LmKr-h1* (*dsLmKr-h1*) (400 ng dsRNA/locust, diluted in *L. migratoria* Ringer). A second injection was given on day three of the fourth nymphal stage (N4 D3) to ensure an efficient knockdown of *LmKr-h1* during this nymphal stage. As a control condition, another group of 183 locusts was injected with dsRNA against *GFP* (*dsGFP*) following the same injection scheme. In our *L. migratoria* colony, we previously observed that the penultimate (N4) and final (N5) nymphal stage are lasting for 5 and 7 days, respectively. Female adult locusts are in the vitellogenic stage of their first gonadotrophic cycle 12 days (D12) after their final molt. As such, several time points during these developmental stages were chosen for tissue collection: day 4 for fourth nymphal female locusts (N4 D4), day 0 (N5 D0), 3 (N5 D3) and 6 (N5 D6) for fifth nymphal control locusts and day 0 (D0), 4 (D4), 8 (D8) and 12 (D12) for the adult (Ad) and adultoid (Ao) locusts.

### 4.6. Observing Ecdysis

The ecdysis of 83 *dsLmKr-h1* and 183 *dsGFP*-injected female locusts was closely observed. Starting from fourth nymphal locusts, synchronized on the day of ecdysis, the duration and type of each developmental stage were monitored in detail. Data were analyzed with a log-rank (Mantel–Cox) test using GraphPad Prism 6 (GraphPad Software Inc., San Diego, CA, USA). 

### 4.7. Measurement of Oocyte Length

Oocytes at the base of the ovarioles, were carefully removed from the dissected ovaries of adult (*dsGFP* control) and adultoid (*dsKr-h1*) female locusts. Oocyte length was measured using millimeter graph paper. For 12 locusts per condition, the average size of three basal oocytes was calculated. Data were analyzed with a nonparametric Mann–Whitney test using GraphPad Prism 6 (GraphPad Software Inc., San Diego, CA, USA).

### 4.8. Microscopy and Histological Analysis

Images of randomly chosen ovaries and ovarioles dissected from adult (*dsGFP* control) and adultoid (*dsLmKr-h1*) females were obtained with a light microscope (Zeiss SteREO Discovery V8, Oberkochen, Germany) equipped with an AxioCam ICc3 camera using the AxioVision 4.7 (Carl Zeiss-Benelux, Oberkochen, Germany). 

Oocyte sections were made according to Billen (2006) [86]. In brief, the dissected oocytes were fixed in 2% glutaraldehyde for 12 to 24 h where after the glutaraldehyde was replaced by sodium cacodylate buffer and postfixed in 2% osmium tetraoxide. Next, the samples were dehydrated in a graded acetone series and embedded in araldite. Semi-thin (1 µm) sections, obtained using a microtome (Leica EM UC6 microtome, Nussloch, Germany), were stained with methylene blue and thionine. The resulting oocyte sections were visualized using a light microscope (Zeiss Axio Imager Z1, Oberkochen, Germany) equipped with an AxioCam MRm camera (1388 × 1040 pixels) and the software program Zen 2012 (Bleu Edition: Carl Zeiss-Benelux, Oberkochen, Germany).

### 4.9. Observation of Mating, Egg Deposition and Hatching

Mating and oviposition events were monitored in adult (*dsGFP* control) and adultoid (*dsLmKr-h1*) female locusts, which were obtained in a follow-up experiment using the same injection scheme as described in §2.5. In this context, the following combinations were studied: 20 control males × control females, 20 control males × adultoid females and 14 adultoid males × adultoid females. On day 4 of the locusts’ adult/adultoid life, the female locusts were separated from the males and transferred to individual cages where they were fed daily with freshly sprouted wheat leaves. On day 5 of the locusts’ adult/adultoid life, one sexually mature virgin male was introduced to each female and mating (i.e., actual connection between the male and female genitals) was monitored. If mating was observed within two hours of the male’s introduction, the male was allowed to stay with the female for 24 h. Pots filled with a slightly humidified sand/turf mixture were supplied to the females to allow oviposition. These pots were checked daily for egg pods. The total number of deposited eggs per egg pod was counted to analyze fecundity. The percentage of hatchlings was determined to analyze fertility. If no mating was observed within two hours of the male’s introduction, the male was removed, and another mating attempt was made the following day using a different virgin male. A log-rank (Mantel–Cox) test in GraphPad Prism 6 (GraphPad Software Inc., San Diego, CA, USA) was used to compare mating and egg deposition between adult and adultoid female locusts. Statistically significant differences in days between mating and egg laying, the number of eggs laid, days between egg laying and hatching and the number and percentage of hatchlings between the different conditions were found via a nonparametric Mann–Whitney test using GraphPad Prism 6 (GraphPad Software Inc., San Diego, CA, USA). 

## 5. Conclusions

In conclusion, our findings show that the knockdown of *LmKr-h1* in the fourth instar resulted in an adultoid phenotype characterized by precocious development of adult features, an accelerated ovarian maturation and reduced mating. Only the few adultoid females capable of mating were also able to deposit their eggs successfully. Although the fecundity of the adultoid females did not seem to differ from that of control adults, the percentage of hatchlings was much lower. When taking into account the major problems that are still caused by migrating locust swarms in many countries worldwide, RNAi-based and/or pharmacological strategies targeting JH signaling pathway components that proved to be crucial for locust development and reproduction would probably be very effective means to control locust population growth, swarm formation and expansion.

## Figures and Tables

**Figure 1 ijms-21-06058-f001:**
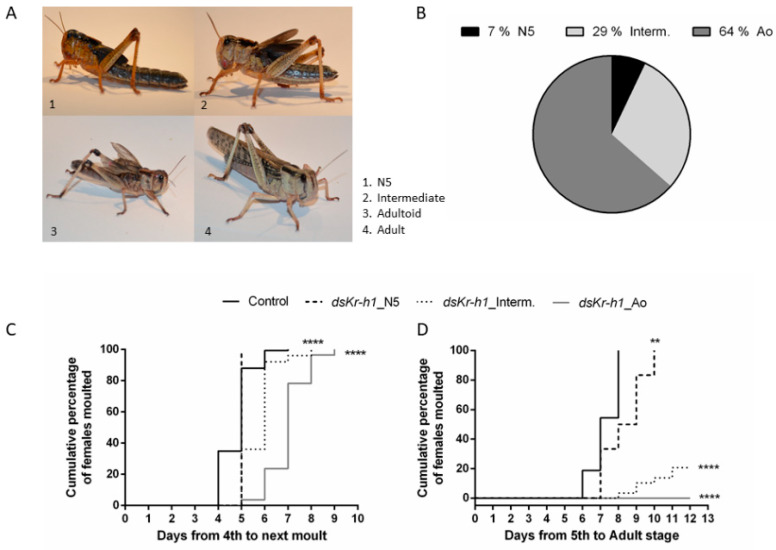
Phenotypes of *dsLmKr-h1*-injected female locusts and the timing of their ecdysis. The timing of ecdysis of 148 *dsGFP* (control) and 83 *dsLmKr-h1*-injected female locusts was observed starting from freshly molted fourth nymphal locusts. (**A**) Observed phenotypes of *dsGFP* (control) and/or *dsLmKr-h1*-injected locusts obtained after molting: (1) fifth nymphal stage (N5), (2) intermediate (Interm.), (3) adultoid (Ao), (4) adult (Ad). (**B**) A pie chart representing the percentages of phenotypes (N5, Interm, Ao) observed after molting of the *dsLmKr-h1-injected* fourth nymphal (N4) locusts. (**C**) The cumulative percentage of females that molted from the fourth nymphal to the fifth nymphal (control and *dsKr-h1*_N5) or to a precocious adult-like stage, classified as an intermediate (*dsKr-h1*_Interm.) or an adultoid phenotype (*dsKr-h1*_Ao). (**D**) The cumulative percentage of females that molted from N5, Interm. or Ao to the adult stage. Statistically significant differences (*p*) between the conditions were found via a log-rank (Mantel–Cox) test and are indicated by asterisks (** *p* < 0.01; **** *p* < 0.0001).

**Figure 2 ijms-21-06058-f002:**
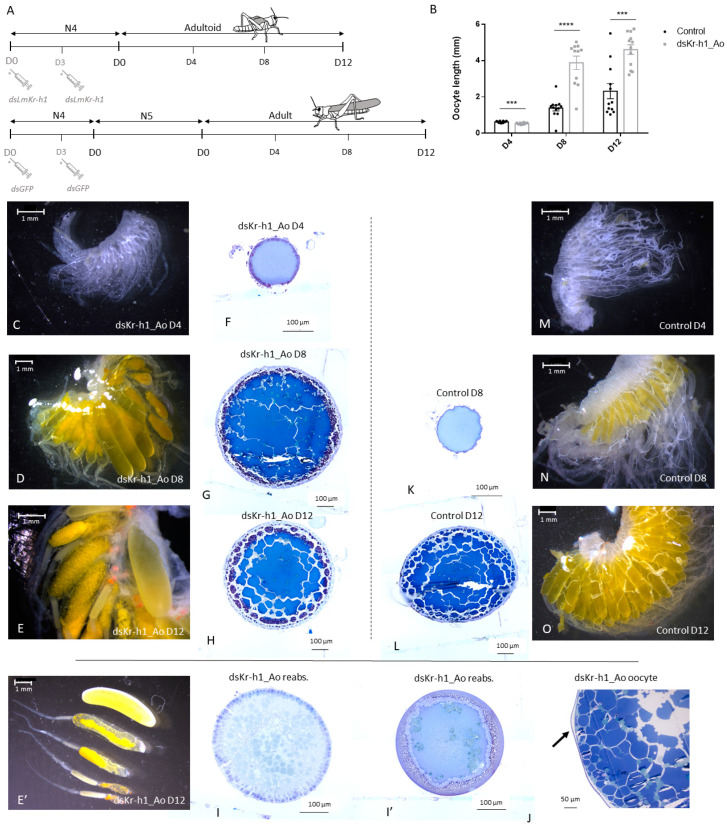
Ovarian maturation in adult and adultoid migratory locusts (*L. migratoria*). (**A**) Time scheme showing the different time points of dsRNA injection together with the duration of each developmental stage and the different time points chosen to dissect control adult (*dsGFP*) and adultoid (*dsLmKr-h1*) female locusts. (**B**) The oocyte length (mm) of adult (control: black bars) and adultoid (*dsKr-h1*_Ao: grey bars) female locusts was measured on day 4 (D4), day 8 (D8) and day 12 (D12) after the final molt. Each point represents the mean length of 3 randomly chosen basal oocytes (mean ± S.E.M). Statistically significant differences (*p*) were found via a nonparametric Mann–Whitney test and are indicated by asterisks (*** *p* < 0.001; **** *p* < 0.0001). (**C**–**E’** and **M**–**O**) Ovaries of 4-day-old (**C**), 8-day-old (**D**) and 12-day-old (**E,E’**) adultoid (*dsKr-h1*_Ao) locusts compared to 4-day-old (**M**), 8-day-old (**N**) and 12-day-old (**O**) adult locusts (control). Scale bars: 1 mm. (**F–L**) Histological sections of basal oocytes from 4-day-old (**F**), 8-day-old (**G**) and 12-day-old (**H**) adultoid (*dsKr-h1*_Ao) locusts compared to 8-day-old (**K**) and 12-day-old (**L**) adult locusts (control). Scale bar: 100 µm. (**I–J**) Histological sections of reabsorbing oocytes (**I, I’**) and an oocyte (**J**) found in the oviduct of an adultoid locust (*dsKr-h1*_Ao), with arrow (black) indicating the chorion. Scale bars: 100 µm (**I, I’**) and 50 µm (**J**).

**Figure 3 ijms-21-06058-f003:**
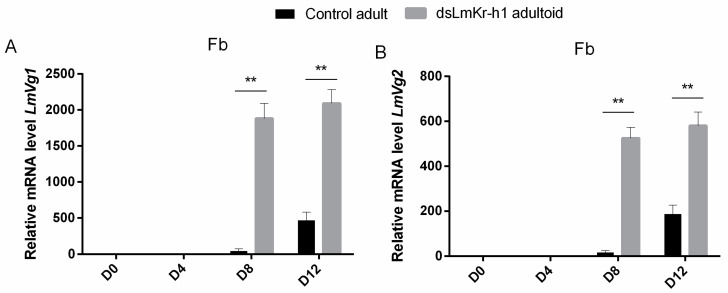
Relative *LmVg1* and *LmVg2* transcript levels in adult and adultoid females. Relative transcript levels of *LmVg1* (**A**) and *LmVg2* (**B**) in the fat body of freshly molted (D0), 4-day-old (D4), 8-day-old (D8) and 12-day-old (D12) adultoid (*dsLmKr-h1*) and adult (*dsGFP*: control) female locusts. The data represent mean ± S.E.M. of four independent pools of three animals, run in duplicate and normalized to *rps13* and *CG13220* transcript levels. Statistically significant differences between the measurements were found via a *t*-test on log-transformed data (with or without two-sided Welch’s correction) and are indicated by asterisks (** *p* < 0.01).

**Figure 4 ijms-21-06058-f004:**
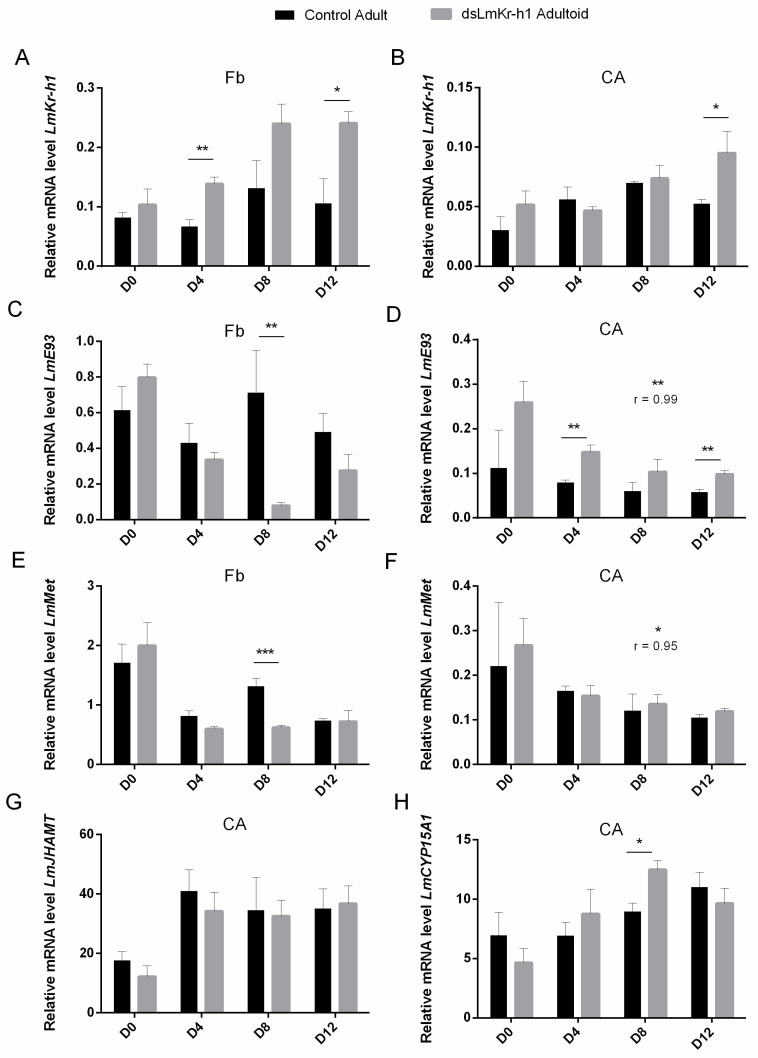
Relative transcript levels of MEKRE93 and JH biosynthesis pathway components in adult and adultoid *L. migratoria* females. Relative *LmKr-h1* (**A + B**)*, LmE93* (**C + D**) and *LmMet* (**E + F**) transcript levels were measured in the fat body (**A + C + D**) and the CA (**B + D + F**) of freshly molted (D0), 4-day-old (D4), 8-day-old (D8) and 12-day-old (D12) adultoid (*dsLmKr-h1*) and adult (*dsGFP*: control) female locusts. Relative *LmJHAMT* (**G**) and *LmCYP15A1* (**H**) transcript levels were measured in the CA as well. The data represent mean ± S.E.M. of four independent pools of three animals, run in duplicate and normalized to *rps13* + *CG13220* and *rp49* + *rps13* transcript levels for fat body and CA samples, respectively. Statistically significant differences between the measurements were found via a *t*-test on log-transformed data (with or without two-sided Welch’s correction) and are indicated by (an) asterisk(s) (* *p* < 0.05; ** *p* < 0.01; *** *p* < 0.001). The correlation coefficient (r) between the transcript profiles of adult and adultoid locusts was obtained by a Pearson correlation calculation. This coefficient is indicated above the figure panels where a significant correlation was observed [*p*-value is indicated by (an) asterisk(s) (* *p* < 0.05; ** *p* < 0.01)].

**Figure 5 ijms-21-06058-f005:**
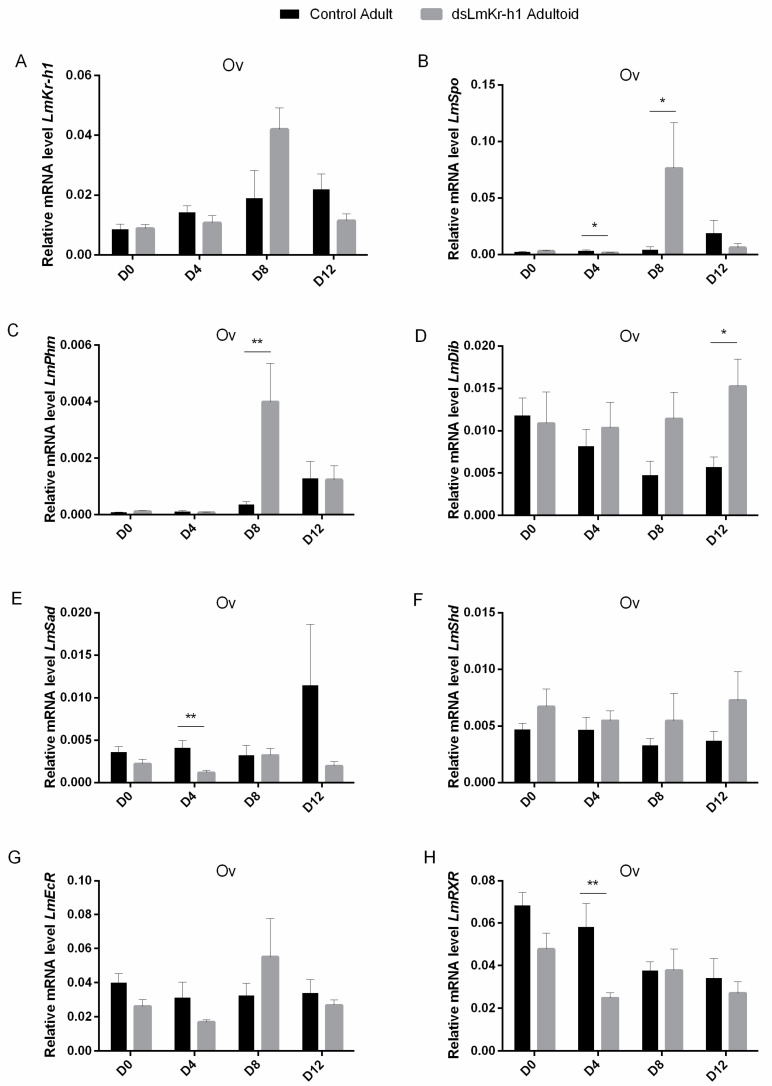
Relative transcript levels of *LmKr-h1*, several *Halloween* genes as well as *LmEcR and LmRXR* in ovaries of adult and adultoid *L. migratoria* females. Relative *LmKr-h1* (**A**)*, Spook* (*LmSpo*) (**B**)*, Phantom* (*LmPhm*) (**C**), *Disembodied* (*LmDib*) (**D**), *Shadow* (*LmSad*) (**E**), *Shade* (*LmShd*), (**F**)*, LmEcR* (**G**) and *LmRXR* (**H**) transcript levels were measured in the ovaries of freshly molted (D0), 4-day-old (D4), 8-day-old (D8) and 12-day-old (D12) adultoid (*dsLmKr-h1*) and adult (*dsGFP*: control) female locusts. The data represent mean ± S.E.M. of four independent pools of three animals, run in duplicate and normalized to *CG13220* and *TubA1* transcript levels. Statistically significant differences between the measurements were found via a *t*-test on log-transformed data (with or without two-sided Welch’s correction) and are indicated by (an) asterisk(s) (* *p* < 0.05; ** *p* < 0.01). With a Pearson correlation calculation, no significant correlations were found between transcript profiles of adult and adultoid locusts.

**Figure 6 ijms-21-06058-f006:**
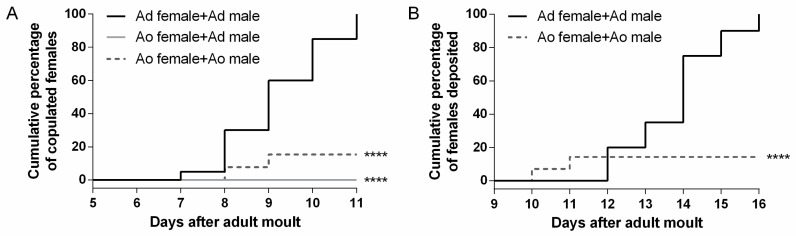
Mating and oviposition by adult and adultoid *L. migratoria* females. Mating and oviposition were observed for 20 adult (Ad) females (each combined with an adult male: ‘Ad female + Ad male’), 20 adultoid (Ao) (each combined with an adult male: ‘Ao female + Ad male’) and 14 Ao (each combined with an adultoid male: ‘Ao female + Ao male’) females starting on day 5 of the adult (Ad) or adultoid (Ao) stage. (**A**) The cumulative percentage of mating females (achieving an actual connection between male and female genitals) and (**B**) the cumulative percentage of females that deposited eggs are shown. Since no mating was observed between adultoid females and adult males, the observation of oviposition was only made for the control condition, combining an adult female with an adult male, and the experimental condition, combining an adultoid female with an adultoid male. Statistically significant differences (*p*) between the control and experimental conditions were found via a log-rank (Mantel–Cox) test and are indicated by asterisks (**** *p* < 0.0001).

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
