# Peer review of "Precocious Downregulation of Krüppel-Homolog 1 in the Migratory Locust, Locusta migratoria, Gives Rise to An Adultoid Phenotype with Accelerated Ovarian Development but Disturbed Mating and Oviposition"

_ijms, 2020, doi:10.3390/ijms21176058_

Round 1

Reviewer 1 Report

The manuscript of Gijbels et. al. reports that precocious downregulation of Krüppel-homolog 1 in the migratory locust results in an adultoid phenotype with accelerated ovarian development. The manuscript is well written and the experiments are clear with an adequate methodology and correct statistical analysis. The figures are easy to follow and the results are well discussed. The MS is a good contribution to further understand the effects of JH signaling manipulation on development and reproduction in Holometabolan insects.

Author Response

We appreciate these positive review comments.

Reviewer 2 Report

This manuscript describes the effects of Kruppel-homolog RNAi in both penultimate (4th) and last (5th) instar nymphal locusts with the former causing the formation of precocious adults that they call “adultoids” whereas the latter does not disturb metamorphosis.  They find that the adultoids spend two days longer in the 4th instar, then molt to precocious adults the size of a 5th instar nymph.  These adultoids mature eggs precociously (8 days after emergence rather than 12 days for normal adults) and can mate with male adultoids but not with normal size male adults. After mating, fewer eggs were oviposited, possibly because of their size which was the same as that of the normal adult (none were laid by virgin females), but very few hatched.  They also compared the levels of mRNA for E93, the adult-specific transcription factor, and for the enzymes of the ecdysteroid and juvenile hormone (JH) biosynthesis pathways in the fat body, the corpora allata and the ovaries in the adultoids to that of normal adults.

              This study is comprehensive, well done, and the results presented well.  The conclusions made are well supported by the data. The statistical analysis is good.

              Several minor points need attention before publication:

1) It would be better to use “precocious adult” rather than “adultoid”.  In the insect endocrinological literature, “adultoid” is usually used for nymphal-adult intermediates formed by administration of juvenile hormone or its mimics to final instar nymphs.

2) The Results section could be more succinct by letting the figures speak for themselves more and hitting the major highlights in the text.

3) lines 222-23:  When comparing ages of adults and precocious adults, it is only appropriate to use time after emergence to that stage.  The time before metamorphosis occurs is not relevant to reproductive maturation that occurs after adult emergence.

4) Since Kr-h1 mRNA levels rise faster in the “adultoids” than in normal adults and the adultoids are created by Kr-h1 RNAi, it would be good to have a supplementary figure as Figure S1b that showed the duration of the knockdown of Kr-h1 in these 4th instar nymphs that become adultoids.  I am assuming that the RNAi is no longer effective after adultoid emergence, but it would be good to verify that.

5) line 291: …profiles was not observed….

6) Figure 4:  One should also do the statistical analysis for the developmental time course of  each of the mRNAs in the control adults versus the adultoids in addition to the statistical analysis at each time point. A similar analysis should be done for the ovary in Fig. 5.

7) lines 555-557: You should dissect some testes of adultoid males to see if there is sperm and also dissect the spermatheca in the female to see if sperm is transferred.  These re simple experiments to answer the question posed here and will make the description of these adultoids more comprehensive.

8) In the References, be sure to use Sentence Case in all the titles of the articles.  Reference #s 18,20, 31, 45, 51, 52, and 74 need to be corrected.

Author Response

We are very grateful to the reviewer for taking the time to read our manuscript and for sharing his/her relevant remarks and constructive suggestions. A point-by-point response to the reviewer’s comments is given below. We have addressed all the listed remarks/questions and revised our manuscript accordingly, as further explained below:

… This study is comprehensive, well done, and the results presented well. The conclusions made are well supported by the data. The statistical analysis is good.

We appreciate this positive feedback and thank the reviewer for the thorough review and detailed comments. The following were listed by the reviewer as minor points to be addressed:

Point 1: It would be better to use “precocious adult” rather than “adultoid”. In the insect endocrinological literature, “adultoid” is usually used for nymphal-adult intermediates formed by administration of juvenile hormone or its mimics to final instar nymphs.

Response 1: We appreciate the reviewer’s comment as the “adultoids” described in our manuscript can indeed be considered as “precocious adults” or perhaps even more correctly as “locusts with a precocious adult-like phenotype”. The term “adultoids” has previously been used in insect endocrinological literature to designate nymphal-adult intermediates formed by administration of juvenile hormone or its mimics to final instar nymphs. However, when browsing through recent literature that reports on the role of Kr-h1 in postembryonic development of insects, it is clear that “adultoid” was also used for nymphal-adult intermediates formed by administration of dsRNA against Kr-h1 [1–3]. Moreover, “adultoid” was also used in the case of the precocious Schistocerca gregaria adults that were observed after administration of dsRNA against Spo [4]. Given that “adultoid” literally means “adult-like”, that several online dictionaries also indicate this as meaning (“a premature adult form of an insect”), and that we clearly explained in our manuscript what we exactly mean with it, we prefer to keep the name because we feel that it cannot be wrongly interpreted in the context of our manuscript. 

Point 2: The Results section could be more succinct by letting the figures speak for themselves more and hitting the major highlights in the text.

Response 2: We thank the reviewer for this suggestion and are very pleased to read that our figures can speak for themselves. Nevertheless, we prefer to also describe the results in sufficient detail in the text, in which we always clearly refer to the corresponding (sub)figures.

Point 3: lines 222-23: When comparing ages of adults and precocious adults, it is only appropriate to use time after emergence to that stage. The time before metamorphosis occurs is not relevant to reproductive maturation that occurs after adult emergence.

Response 3: We agree with the statement made by the reviewer. Our data comparison of normal and precocious (adultoid) adults was always based on the timing after adult emergence age. In lines 219-222 (Revised version), we clearly use time after emergence to adult/adultoid when referring to the significant effects on ovarian maturation. The sentence (original manuscript lines 222-223) just reminds the reader to the fact that the adultoid female locusts, which are characterised by an accelerated ovarian maturation, are actually also several days younger than the control female locusts when considering their total age since hatching (Revised version lines 222-224). We slightly adapted the sentence to avoid possible misinterpretation. This now reads as follows “Moreover, considering that a fifth nymphal stage preceded their adult moult, as illustrated in Figure 2A, the locusts of the control group (dsGFP) were in total at least four days older than the adultoid (dsLmKr-h1) females”.

Point 4: Since Kr-h1 mRNA levels rise faster in the “adultoids” than in normal adults and the adultoids are created by Kr-h1 RNAi, it would be good to have a supplementary figure as Figure S1b that showed the duration of the knockdown of Kr-h1 in these 4th instar nymphs that become adultoids. I am assuming that the RNAi is no longer effective after adultoid emergence, but it would be good to verify that.

Response 4: The main focus of our manuscript is on the reproductive physiology of adultoids. RNAi of LmKr-h1 has been employed (with only two injections in the first half of the fourth nymphal stage, on N4 D0 and N4 D3) as a tool to produce these adultoids. As described in our manuscript, a temporary knockdown effect was observed on LmKr-h1 expression levels in fat body on N4 D4, as well as precociously increased LmE93 levels. These data are shown in Suppl. Fig S3. Indeed on the day of the adultoid moult (Ao D0), the LmKr-h1 levels were clearly much higher than in the dsGFP injected (control) N5 condition. Therefore, the effect of the knockdown was temporally restricted (and this temporal effect had important developmental consequences, as described in the manuscript). To stress this fact more clearly in the manuscript, we added the following sentence to the Discussion section (Revised version lines 409-412): “On the day of the adultoid moult (Ao D0), the LmKr-h1 levels were clearly much higher than in the dsGFP injected (control) N5 condition and resembled the normal adult levels (Ad D0) (Suppl. Fig S3A), suggesting that the effect of the knockdown was temporally restricted to the N4 stage.”

Point 5: line 291: …profiles was not observed….

Response 5: We apologize for this mistake. The sentence has been corrected according to the reviewer’s remark (Revised version line 293).

Point 6: Figure 4: One should also do the statistical analysis for the developmental time course of each of the mRNAs in the control adults versus the adultoids in addition to the statistical analysis at each time point. A similar analysis should be done for the ovary in Fig. 5.

Response 6: We thank the reviewer for this keen remark and have added the results obtained from a Pearson correlation calculation (when significant). These results showed that the temporal profiles of the relative mRNA levels of LmE93 and LmMet, measured in the CA of control (dsGFP) adult females, correlate well with these of the adultoid (dsKr-h1) females. This information was added to the manuscript and in corresponding figure legends:

-Results section (lines 298-300): “In a Pearson correlation calculation, the temporal profiles of the relative mRNA levels of LmE93 and LmMet in the CA of control (dsGFP) adult females correlated well with these of adultoid (dsLmKr-h1) females (Fig. 4D and F).”

-Legend of Fig. 4 (lines 316-319): “The correlation coefficient (r) between the transcript profiles of adult and adultoid locusts was obtained by a Pearson correlation calculation. This coefficient is indicated above the figure panels where a significant correlation was observed [p-value is indicated by (an) asterisk(s) (* p < 0.05; ** p < 0.01)].”

-Legend of Fig. 5 (lines 339-340): “With a Pearson correlation calculation no significant correlations were found between transcript profiles of adult and adultoid locusts.”

-Results section (lines 345-347): “No correlations were found between adult and adultoid locusts for the temporal profiles of ovarian LmKr-h1, LmSpo, LmPhm, LmDib, LmSad, LmShd, LmEcR and LmRXR transcripts.”

-Material and Method section (lines 642-643): “The correlation coefficient (r) between the transcript profiles of adult and adultoid locusts was obtained by a Pearson correlation calculation.”

Point 7: lines 555-557: You should dissect some testes of adultoid males to see if there is sperm and also dissect the spermatheca in the female to see if sperm is transferred. These re simple experiments to answer the question posed here and will make the description of these adultoids more comprehensive.

Response 7: This is an interesting suggestion. However, we did not check for all this, since only two females had mated and we did not want to sacrifice these to check for presence of sperm in their spermatheca. This would not yet have been the full proof of a successful fertilization resulting in viable offspring. Therefore, our choice was to let them deposit their eggs and verify whether this would lead to hatchlings. Since it is good suggestion, we have decided to add one more sentence (Revised version lines 584-586) in the Discussion section, to emphasize the importance of further investigating the availability of sperm from adultoid males: “Future dissections of adultoid male testes and adultoid female spermatheca, to verify the presence and transfer of sperm, would probably enable a more comprehensive interpretation of these data.”

Point 8: In the References, be sure to use Sentence Case in all the titles of the articles. Reference #s 18,20, 31, 45, 51, 52, and 74 need to be corrected.

Response 8: We would like to apologize for these inconsistencies. The references have now been adapted according to the reviewers’ and editor’s remarks.

References (cf. ‘Response 1’):

[1]      Li KL, Yuan SY, Nanda S, et al. The Roles of E93 and Kr-h1 in Metamorphosis of Nilaparvata lugens. Front Physiol 2018; 9: 1677.

[2]      Konopova B, Smykal V, Jindra M. Common and distinct roles of juvenile hormone signaling genes in metamorphosis of holometabolous and hemimetabolous insects. PLoS One 2011; 6: e28728.

[3]      Smykal V, Daimon T, Kayukawa T, et al. Importance of juvenile hormone signaling arises with competence of insect larvae to metamorphose. Dev Biol 2014; 390: 221–230.

[4]      Sugahara R, Tanaka S, Shiotsuki T. RNAi-mediated knockdown of SPOOK reduces ecdysteroid titers and causes precocious metamorphosis in the desert locust Schistocerca gregaria. Dev Biol 2017; 429: 71–80.

Reviewer 3 Report

The manuscript entitled “Precocious downregulation of Krüppel-homolog 1 in the migratory locust, Locusta migratoria, gives rise to an adultoid phenotype with accelerated ovarian development” present very important and interesting results concern role of JH signalling during the insects development and maturation of reproductive system. Generally, the manuscript is well-written and the research are well-designed. Also, the great value of presented research is it complexity. I have only few minor remarks which should be clarified, before potential manuscript acceptance.

Minor remarks:

  1. Results section. In my opinion the value of statistical tests should be also added in the manuscript text, especially when the Authors mentioned about significant differences between tested groups.
  2. Materials and Method section.

Subsection “4.1. Rearing of animals”. Tested individuals were kept at an ambient relative humidity between 40 % and 60 %. This indicate on unstable rearing conditions, especially in fact that insects development and also ovary maturation can be correlated with humidity (Broufas et al., 2009; Lu and Wu, 2011; Yang et al., 2015). Similar dependencies we can observed across different Insect orders, including Orthoptera (Ali, 1982; Brett, 2018). Of course, some of the presented results are clear and differences are significant, but in some cases (some of the PCR analysis) the variability of results is quite large, which can be related to different rearing conditions.

Subsection “4.7. Measurement of oocyte length” the Authors mentioned that “Oocyte length was measured using millimetre graph paper.”. Due to this I worry about accuracy of measurements. It is intriguing why the Authors did not use one of the microscopic software which allow a quick and easy measurement?

The Authors also should in Material and Method section add information about number of individuals used in each of assay.

  1. Please carefully check whether each species names and some Latin names (like corpora allata) are italicized.

References

Ali, S. 1982. Effect of temperature and humidity on the development and fertility-fecundity of Aerida exaltata Walk. Proceedings: Animal Sciences 91:267-273.

Brett, C.H. 2018. Interrelated Effects of Food, Temperature, and Humidity on the Development of the Lesser Migratory Grasshopper,'Melanoplus mexicanus Mexicanus'(Saussure)(Orthoptera).

Broufas, G., M. Pappas, and D. Koveos. 2009. Effect of relative humidity on longevity, ovarian maturation, and egg production in the olive fruit fly (Diptera: Tephritidae). Ann Entomol Soc Am 102:70-75.

Lu, Y., and K. Wu. 2011. Effect of relative humidity on population growth of Apolygus lucorum (Heteroptera: Miridae). Appl Entomol Zool 46:421-427.

Yang, Y., W. Li, W. Xie, Q. Wu, B. Xu, S. Wang, C. Li, and Y. Zhang. 2015. Development of Bradysiaodoriphaga (Diptera: Sciaridae) as affected by humidity: an age–stage, two-sex, life-table study. Appl Entomol Zool 50:3-10.

Author Response

We are very grateful to the reviewer for taking the time to read our manuscript and for sharing his/her relevant remarks and constructive suggestions. A point-by-point response to the reviewer’s comments is given below. We have addressed all the listed remarks/questions and revised our manuscript accordingly, as further explained below:

… Generally, the manuscript is well-written and the research are well-designed. Also, the great value of presented research is it complexity. I have only few minor remarks which should be clarified, before potential manuscript acceptance.

We appreciate this positive feedback and thank the reviewer for the thorough review and detailed comments.

Point 1: Results section. In my opinion the value of statistical tests should be also added in the manuscript text, especially when the Authors mentioned about significant differences between tested groups.

Response 1: We thank the reviewer for this suggestion, but we believe that referring to the figures (which, according to another reviewer, may actually ‘speak for themselves’) where the different significance levels have been clearly indicated by asterisks (p-value levels are indicated in the figure legends) should be sufficient.

Point 2: Materials and Method section. Subsection “4.1. Rearing of animals”. Tested individuals were kept at an ambient relative humidity between 40 % and 60 %. This indicate on unstable rearing conditions, especially in fact that insects development and also ovary maturation can be correlated with humidity (Broufas et al., 2009; Lu and Wu, 2011; Yang et al., 2015). Similar dependencies we can observed across different Insect orders, including Orthoptera (Ali, 1982; Brett, 2018). Of course, some of the presented results are clear and differences are significant, but in some cases (some of the PCR analysis) the variability of results is quite large, which can be related to different rearing conditions.

Response 2: We appreciate the comment and understand the concern of the reviewer as insect development and also ovary maturation can be correlated with humidity [1–5]. However, the cages containing control (dsGFP) and experimental (dsLmKr-h1) groups of locusts in our study were always kept simultaneously in the same room to make sure that they also experienced the same conditions for development and ovary maturation. We believe that variation is an intrinsic characteristic of most, if not all, (developing/evolving) biological systems and cannot be completely eliminated.  

Point 3: Subsection “4.7. Measurement of oocyte length” the Authors mentioned that “Oocyte length was measured using millimetre graph paper.”. Due to this I worry about accuracy of measurements. It is intriguing why the Authors did not use one of the microscopic software which allow a quick and easy measurement?

Response 3: We appreciate these concerns of the reviewer, but we believe that measuring the oocyte length using millimetre graph paper is sufficiently accurate. By using a binocular microscope, lengths differing 0.2 mm are distinguishable. Anyway, one does not need an accuracy at nm or µm level, if standard deviations are not in this order of magnitude. In addition, for each locust the average size of three basal oocytes was calculated and each condition consisted of 12 locusts. Moreover, the differences in length (Fig. 2) observed between adult and adultoid females were statistically very significant.

Point 4: The Authors also should in Material and Method section add information about number of individuals used in each of assay.

Response 4: We followed the advice of the reviewer. Extra information about the numbers of individuals used in each assay is now included (Revised version lines 659, 664, 672, 679, 683, 697 and 698).  

Point 5: Please carefully check whether each species names and some Latin names (like corpora allata) are italicized.

Response 5: We thank the reviewer for this comment. By carefully checking the manuscript, one species name (Revised version line 590) and two Latin names (Revised version lines 25 and 39) were adapted accordingly.

References (cf. ‘Point 2’):

[1]      Broufas GD, Pappas ML, Koveos DS. Effect of relative humidity on longevity, ovarian maturation, and egg production in the olive fruit fly (Diptera: Tephritidae). Ann Entomol Soc Am 2009; 102: 70–75.

[2]      Lu Y, Wu K. Effect of relative humidity on population growth of Apolygus lucorum (Heteroptera: Miridae). Appl Entomol Zool 2011; 46: 421–427.

[3]      Yang Y, Li W, Xie W, et al. Development of Bradysiaodoriphaga (Diptera: Sciaridae) as affected by humidity: an age–stage, two-sex, life-table study. Appl Entomol Zool 2015; 50: 3–10.

[4]      Ali S. Effect of temperature and humidity on the development and fertility-fecundity of Aerida exaltata Walk. Proc Anim Sci 1982; 91: 267–273.

[5]      Brett CH. Interrelated effects of food, temperature, and humidity on the development of the lesser migratory grasshopper Melanoplus Mexicanus Mexicanus (Saussure) (Orthoptera). In: Oklahoma Agricultural Experiment Station. 2018, p. 50.